# Comparison of Two 2.45 GHz Microwave Ablation Devices with Respect to Ablation Zone Volume in Relation to Applied Energy in Patients with Malignant Liver Tumours

**DOI:** 10.3390/cancers14225570

**Published:** 2022-11-13

**Authors:** Simeon J. S. Ruiter, Jamila E. de Jong, Jan Pieter Pennings, Robbert J. de Haas, Koert P. de Jong

**Affiliations:** 1Department of Hepato-Pancreato-Biliary Surgery and Liver Transplantation, University Medical Center Groningen, University of Groningen, 9713 GZ Groningen, The Netherlands; 2Department of Radiology, University Medical Center Groningen, University of Groningen, 9713 GZ Groningen, The Netherlands

**Keywords:** colorectal liver metastasis, hepatocellular carcinoma, thermal ablation, microwave ablation

## Abstract

**Simple Summary:**

Ablation zone volumes (AZV) after microwave ablation (MWA) of malignant liver tumours remain highly unpredictable. The aims of the present study were to compare two 2.45 GHz MWA devices with respect to AZV in relation to the applied energy after MWA in patients with hepatocellular carcinoma or colorectal liver metastasis, and to identify potential confounders for this relationship. We confirmed the unpredictability of AZVs based on the applied output energy for both tumour types. We observed no differences in the ratio between AZV and the applied energy between tumour types. The ratio between AZV and applied energy was different between the two MWA devices; however, its reflected energy due to differences in cable and antenna design remains unclear and might contribute to these differences.

**Abstract:**

Purpose: (i) to compare two 2.45 GHz MWA devices with respect to AZV in relation to the applied energy after MWA in patients with hepatocellular carcinoma (HCC) or colorectal liver metastasis (CRLM) and (ii) to identify potential confounders for this relationship. Methods: In total, 102 tumours, 65 CRLM and 37 HCC were included in this retrospective analysis. Tumours were treated with Emprint (*n* = 71) or Neuwave (*n* = 31) MWA devices. Ablation treatment setting were recorded and applied energy was calculated. AZV and tumour volumes were segmented on the contrast-enhanced CT scans obtained 1 week after treatment. The AZV to applied energy *R*(AZV:E) ratios were calculated for each tumour treatment and compared between both MWA devices and tumour types. Results: *R*(AZV:E)_EMPRINT_ was 0.41 and *R*(AZV:E)_NEUWAVE_ was 0.81, *p* < 0.001. Moderate correlation between AZV and applied energy was found for Emprint (r = 0.57, *R*^2^ = 0.32, *p* < 0.001) and strong correlation was found for Neuwave (r = 0.78, *R*^2^ = 0.61, *p* < 0.001). *R*(AZV:E)_CRLM_ was 0.45 and *R*(AZV:E)_HCC_ was 0.52, *p* = 0.270. Conclusion: This study confirms the unpredictability of AZVs based on the applied output energy for HCC and CRLM. No significant differences in *R*(AZV:E) were observed between CRLM and HCC. Significantly lower *R*(AZV:E) was found for Emprint devices compared to Neuwave; however, reflected energy due to cable and antenna design remains unclear and might contribute to these differences.

## 1. Introduction

Microwave ablation (MWA) is an established tissue-sparing treatment for malignant liver tumours such as hepatocellular carcinoma (HCC) and colorectal liver metastases (CRLM) [1,2]. In non-randomized studies, hepatic resection and MWA yield similar treatment efficacy and overall survival for CRLM [3,4,5,6,7,8]. Additionally, thermal ablation is an effective and repeatable therapy for primary and recurrent HCC [9].

The most important drawback of thermal ablation is the appearance of viable tumour tissue at the edge of the ablation zone [ablation site recurrence (ASR)] [10,11]. Complete tumour coverage with an adequate ablation margin assessed in 3D is crucial for treatment success [12,13,14,15]. To achieve complete tumour coverage, creation of predictable ablation volumes is essential and depends, among other things, on the applied energy controlled by the power and time setting of the ablation device. Ablation protocols provided by ablation device manufacturers are mostly based on experiments with ex vivo and non-perfused animal livers with normal liver parenchyma [16]. Scarce clinical reporting on applied energy and ablation volumes suggests that liver parenchyma characteristics such as perfusion (perivascular tumour location) and tissue properties (fatty or cirrhotic liver disease) might influence the conduction of MWA energy and thus have an impact on the size and the shape of ablation zone volumes (AZV) [17,18]. Previous studies showed that besides the applied energy, other clinical factors, including physical properties of tumour and liver tissue, might affect AZV, resulting in poor predictability [19,20]. However, none of these studies compared and reported outcomes using the Neuwave MWA device.

The aims of the present study were (i) to compare two 2.45 GHz MWA devices with respect to AZV in relation to the applied energy after MWA in patients with HCC and CRLM and (ii) to identify potential confounders for this relationship.

## 2. Materials and Methods

### 2.1. Patient Population

In the present study, we analysed patients who underwent percutaneous CT-guided MWA for either CRLM or HCC in the period between January 2017 and June 2021. All patients were discussed in a multidisciplinary tumour board meeting in which the decision for CT guided MWA was made. Patients were included in this study if (1) they were treated for HCC or CRLM and (2) they were treated with either an Emprint device (Emprint MWA, COVIDien/Medtronic, Minneapolis, MN, USA) or a Neuwave device (Neuwave MWA, Ethicon, Madison, WI, USA). The decision on which device was used depended on the operator’s choice. Tumours were excluded if (1) they were previously treated by MWA [ablation site recurrence (ASR), (2) there was an inability to separate the AZV of multiple tumours located close to each other, (3) they were missing contrast-enhanced CT scans (CE-CT), (4) there was an unclear demarcation of the tumour on CE-CT, (5) there was missing energy data, and (6) simultaneously multi-probe ablations were also excluded. Ethical approval was obtained by the Institutional Review Board of the University Medical Center Groningen, and the need for informed consent was waived.

### 2.2. Procedures

All procedures were performed in the interventional CT suite by interdisciplinary teams consisting of specifically trained interventional radiologists and hepatobiliary surgeons with extensive experience in image-guided tumour ablation (all operators had performed >100 MWA procedures). Patients were placed under general anaesthesia and positioned on a vacuum mattress to eliminate patient movement. Controlled apnoea was applied during image acquisition and antenna manipulations. Procedures were performed with 64-multidetector row CT systems (Siemens Somatom Sensation 64). All CE-CT scans had an in-plane resolution of 0.6–1.0 mm × 0.6–1.0 mm and a slice thickness of 1 mm. A CE-CT scan was performed in arterial and/or portal venous phase for the planning of ablation trajectories before antenna placement. A non-enhanced CT scan was acquired after each antenna manipulation for validation of antenna positions. Single-antenna MWA was performed with the Emprint or Neuwave device. Appropriate time and power settings for MWA cycles were chosen according to the protocol provided by the manufacturer with modifications according to the judgment of the team. Larger lesions were treated by multiple antenna positions, creating partially overlapping ablation zones. A confirmation CE-CT scan was performed in the arterial and portal venous phase directly after antenna extraction, and validation of technical success was evaluated by direct overlay of co-registered pre- and post-ablation CE-CT images. Technical success was defined as complete coverage of the tumour by the AZV including an ablation margin of >5 mm accordingly to standardized terminology and reporting criteria [21]. If this was not achieved, immediate re-ablation was performed. When kidney function allowed, a second confirmation CE-CT was performed after the re-ablation. Control CE-CT scans were acquired 5–7 days after the ablation procedure.

### 2.3. Data Extraction and Analysis

Patient and tumour characteristics, details on neoadjuvant chemotherapy, ablation time, ablation power and number of antenna positions were collected in the study database. A subcapsular tumour location was defined as a tumour within 5 mm of the liver capsule. A perivascular location was defined as a tumour within 5 mm to a vessel with a diameter of ≥3 mm. Applied energy (E) was calculated by:(1)E (kJ)=power(W)∗time(s)/1000
to determine the AZV and tumour volumes, pre- and post-CE-CT ablation scans were analysed retrospectively. Ablation zone and tumour volumes were segmented using semi-automatic segmentation tools available in High Precision 3D ablation software (Philips, Eindhoven, the Netherlands). *R*(AZV:E) was determined by dividing the ablation zone volume (AZV) in millilitres by the applied energy (E) in kilojoules for each ablated tumour. *R*(T:AZV) was determined by dividing the tumour volume (T) in millilitres by AZV in millilitres.

### 2.4. Liver Steatosis

Details on liver tissue category (normal, steatotic or cirrhotic) were extracted from pathology reports (obtained from biopsies during MWA procedure) or radiology data when no histology was available. As described before, liver steatosis affects heat conduction properties during MWA [22]. Kim et al. explored the possibilities of CE-CT for the examination of steatosis and results showed a similar-to-greater accuracy compared to non-enhanced CT scans [23]. As a surrogate marker for steatosis, we used this method for blood-subtracted hepatic attenuation on contrast-enhanced CT. The one-week post-ablation CE-CT scan including both an arterial and portal-veinous phase were used to select regions of interest (ROI) in the abdominal aorta (3 ROIs in the aortic segment between coeliac artery and superior mesenteric artery), portal vein (3 ROIs in main branch), left liver lobe (3 ROIs) and right liver lobe (5 ROIs). The comparison between the aorta (*A*), portal vein (*P*) and liver (*L*) was calculated by:(2)$$HU_{liver}=\frac{L-0.3*\left(0.75P+0.25A\right)}{0.7}$$
as described by Kim et al. [22] The equation takes into account the ratio between the sinusoids (30%) and parenchyma (70%) in the liver and the blood supply by the portal vein (75%) and hepatic artery (25%).

### 2.5. Statistic

Means and standard deviation or median and interquartile range (IQR) were reported for continuous variables and number and percentage for categorical variables. Linear correlation analysis was applied to investigate relationships between AZV and applied energy. Correlations were classified as weak (<0.4), moderate (0.4–0.7) and strong (>0.7) [24]. The chi-square test was applied to compare categorical variables and Mann–Whitney U test for nonparametric continuous variables. The threshold for statistical significance was set at *p* < 0.05. Statistical analyses were performed using IBM SPSS Statistics version 23 (IBM Corporation, New York, NY, USA) and graphs were generated using GraphPad Prisma version 9.1.0 (GraphPad Software, San Diego, CA, USA).

## 3. Results

### 3.1. Patient and Tumour Characteristics

From a total of 212 treated tumours in 136 patients, 102 tumours in 81 patients were eligible for enrolment in the current volumetric analyses. Figure 1 shows the reasons for exclusion. Clinicopathological characteristics of tumours and patients included in the study are shown in Table 1. Median tumour diameter was 19 mm (IQR 14–27 mm, range 5–54 mm) and median tumour volume was 2.92 mL (IQR 1.06–8.00 mL, range 0.13–43.86 mL). In total, 65 CRLM were located in normal or steatotic livers and 37 HCC were predominantly located in cirrhotic livers. Sixteen patients with CRLM received chemotherapy as a neoadjuvant treatment preceding the ablation procedure. Eight patients with CRLM did receive chemotherapy before as a prior treatment, but not in the 6 months before MWA. None of the patients with HCC received chemotherapy prior to MWA. All ablations were performed with a single antenna with a median total applied energy of 84.3 kJ (IQR 56.4–140.0 kJ), and the median number of antenna positions was 3 (IQR 2–4). Fourteen ablations (13.7%) were performed with only one antenna position for the complete treatment of the tumour. Median AZV for all tumours was 41.5 mL (IQR 22.0–61.0 mL). Median ratio between tumour and AZV (*R*(T:AZV)) was 0.07 (IQR 0.04–0.15). All tumours included in this study were presumed to be treated technically successful, with no residual tumours detected at the end of the ablation procedure, as assessed by direct overlay of co-registered pre- and post-ablation CT images.

### 3.2. Comparison of Devices

In total, 71 tumours were ablated with the Emprint device and 31 tumours with the Neuwave device. Between the Emprint and Neuwave group, significant differences were observed in sex (*p* = 0.029), age (*p* = 0.012), types of underlying liver disease (*p* = 0.003) and tumour types (*p* = 0.001). No significant differences were observed between the groups regarding location, diameter and volume of the tumours. Table 2 shows a summary of ablation outcome parameters per device. The median applied energy (E_EMPRINT_ = 85.5 kJ, E_NEUWAVE_ = 67.2, *p* = 0.062) and median AZV (AZV_EMPRINT_ = 41.9 mL, AZV_NEUWAVE_ = 41.1 mL, *p* = 0.342) were not significantly different between both devices. *R*(AZV:E) was lower for Emprint than Neuwave (*R*(AZV:E)_EMPRINT_ = 0.41, *R*(AZV:E)_NEUWAVE_ = 0.81, *p* < 0.001). Median *R*(T:AZV) was 0.06 for the Emprint, and 0.07 for the Neuwave device, (*p* = 0.513). Figure 2A shows the applied energy in relation to AZV as scatter plots grouped by device. Moderate correlation (r = 0.57, *R*^2^ = 0.32, *p* < 0.001) was found for Emprint and strong correlation (r = 0.78, *R*^2^ = 0.61, *p* < 0.001)was found for Neuwave. Figure 2B shows applied energy in relation to AZV for Emprint comparing CRLM and HCC. Moderate correlations were found for both Emprint-CRLM (r = 0.61, *R*^2^ = 0.37, *p* < 0.001) and Emprint-HCC (r = 0.69, *R*^2^ = 0.48, *p* = 0.004). Strong correlations were found for both Neuwave-CRLM (r = 0.84, *R*^2^ = 0.70, *p* = 0.005) and Neuwave-HCC (r = 0.72, *R*^2^ = 0.52, *p* < 0.001) (Figure 2C).

### 3.3. Comparison of Tumour Types

In total, 65 CRLM and 37 HCC were ablated. The median applied energy (E_CRLM_ = 88.4 kJ, E_HCC_ = 66.0, *p* = 0.109) and median AZV (AZV_CRLM_ = 44.9 mL, AZV_HCC_ = 37.0 mL, *p* = 0.187) were not significantly different between tumour types. No significant differences were found for *R*(AZV:E) between tumour types (*R*(AZV:E)_CRLM_ = 0.45, *R*(AZV:E)_HCC_ = 0.52, *p* = 0.270). Median *R*(T:AZV) was not significantly different between tumour types (*R*(T:AZV)_CRLM_ = 0.06, *R*(T:AZV)_HCC_ = 0.07, *p* = 0.513). Table 3 shows a summary of ablation outcome parameters per device categorized by tumour type. For both devices, no significant differences in R(AZV:E) were observed between tumour types (*p* = 0.398 and *p* = 0.384, respectively). Figure 2D shows the applied energy in relation to the AZV as scatter plots grouped by tumour type. Moderate correlations were found for both CRLM (r = 0.63, *R*^2^ = 0.39, *p* < 0.001) and HCC (r = 0.58, *R*^2^ = 0.34, *p* < 0.001). For patients with CRLM, median *R*(AZV:E) values were not significantly different between tumours with KRAS mutation and KRAS wildtype (0.39 and 0.44, respectively, *p* = 0.885). Median *R*(AZV:E) values of CRLM in patients who did and those who did not receive prior systemic chemotherapy were not significantly different (0.41 and 0.48, respectively, *p* = 0.422). There were no significant differences in median *R*(AZV:E) values between perivascular and non-perivascular tumours (0.46 and 0.49, respectively, *p* = 0.786). Additionally, no significant differences in *R*(AZV:E) were observed between subcapsular and non-subcapsular tumours (0.51 and 0.47, respectively, *p* = 0.779). Significantly different HU_liver_ values were found for normal liver tissue compared to steatotic and cirrhotic liver tissue (88.0, 68.0, and 68.5, respectively, *p* < 0.001). No significant differences in *R*(AZV:E) were found for normal, steatotic and cirrhotic liver parenchyma (0.45, 0.47 and 0.50, respectively, *p* = 0.736). As a result, a weak correlation (r = −0.12, *R*^2^ = 0.02, *p* = 0.224) was found between HU and *R*(AZV:E).

## 4. Discussion

The aim of the present study was to determine the relationship between the applied energy and AZV in patients with HCC and CRLM treated with two different MWA devices. Therefore, *R*(AZV:E) was calculated for each tumour treatment. In this study we found significantly lower *R*(AZV:E) for the Emprint device compared to the Neuwave device. Variability in AZV was explained by the applied energy for 61% for the Neuwave ablations compared to 32% for the Emprint ablations. No differences were found for *R*(AZV:E) between CRLM and HCC (*p* = 0.270). For CRLM, only 39%, and for HCC, only 34% of the variability in AZV could be explained by the applied energy.

Heerink et al. previously investigated the relationship between applied energy and AZV for CRLM and HCC. In contrast to the results of the present study, a significantly higher *R*(AZ:E) for HCC than for CRLM was found [19]. A recent publication by Paolucci et al. reported weak correlations between applied energy and AZV for CRLM [20]. Only tumour radius affected the ratio between applied energy and AZV. In accordance with the findings by Paolucci et al., subcapsular or peri-vascular location and KRAS mutational status of the tumour did not affect *R*(AZV:E).

We found that only 7% (IQR 4–15%) of the AZV consists of tumour tissue; the remaining 93% consists of adjacent liver parenchyma which is in line with the results of Heerink et al. and Paolucci et al. This low *R*(T:AZV) can be explained by the fact that the volume of a spherical tumour of 10 mm (tumour volume of 1 mL) would only be 7% of the AZV when aiming to obtain a tumor-free margin of 10 mm (AZV 14 mL). These findings strongly suggest that surrounding liver parenchyma affects the evolution of the ablation zone significantly more than tumour tissue. However, no significant differences in *R*(AZV:E) were observed between normal, steatotic and cirrhotic liver parenchyma in our analysis. Deshazer et al. already stated that the underlying liver parenchyma (normal vs. cirrhotic) is the main factor influencing the difference in treatment effects between tumour types because HCC are mostly seen in cirrhotic livers, and, on the contrary, CRLM is mostly seen in normal livers [22]. Additionally, a weak correlation (r = 0.12, *R*^2^ = 0.02) was found between liver parenchyma density at CT and *R*(AZV:E). Due to the absence of biopsies of the ablated liver tissue, these results should be interpreted carefully and the relationship between liver parenchyma density at CT, liver steatosis and AZVs remains uncertain.

Clinical data reporting on *R*(AZV:E) are scarce, especially regarding ablation with the Neuwave device [16]. Huber et al. reported a *R*^2^ = 0.41 for the relationship between applied energy and AZV after ablation with Neuwave of several tumour types [25]. Winokur et al. reported outcomes of clinical ablation with Neuwave of several tumour types, but did not report *R*(AZV:E) [26].

Despite the fact that both Emprint and Neuwave devices produce 2.45 GHz output, we found significant differences in the ratio between ablation zone volume and applied energy *R*(AZV:E). For ablations with Neuwave, less energy is needed to create the same ablation zone volume compared to the Emprint device. These differences might be explained by non-identical technical characteristics of both devices. First of all, the Emprint device uses a water-cooling system which might be influenced by the temperature of the water, whereas the Neuwave device uses a CO_2_ cooling system with more stable temperatures. Secondly, antenna and cable designs are different. In addition, Lee and colleagues observed significant differences in sphericity indices and longitudinal tissue contraction between Emprint and Neuwave devices in an ex vivo model [27]. In order to compare both devices more systematically, actually delivered energy in the liver tissue should be measured taking reflected energy into account. Based on the results of this and previous studies, prediction of the size of the ablation zone cannot be done simply on manufacturers’ algorithms generated on the amount of delivered energy from the device. Additionally, these algorithms rather frequently only take into account data of non-human, normal, and non-perfused (non-tumoral) tissue. Therefore, patient- and tumour-tailored monitoring of the ablation zone during the treatment remains crucial. Application of transcatheter CT hepatic arteriography and arterial portography enables repeated contrast-enhanced imaging and nearly real-time monitoring of AZV during the procedure [28].

There are some limitations of the present study. First of all, this study was conducted retrospectively, and potential confounding factors may not be taken into account. Significant differences in clinicopathological characteristics by means of sex, age, type of liver parenchyma, and tumour type were observed between the Emprint and Neuwave group. Additionally, the numbers of included tumours in the HCC and Neuwave groups were small. The majority of ablations in this study were performed with overlapping ablation zones because of multiple overlapping single antenna positions due to tumour size and shape. Overlapping ablations might contribute to differences in *R*(AZV:E) because liver tissue properties will change after ablation. Because of the overlapping ablations it was also not possible to determine the differences in sphericity between both MWA devices. Another potential confounding factor is the absence of histology of the liver parenchyma in 68% of the patients in order to distinguish adequately between normal, steatotic and cirrhotic liver tissue. Lastly, tissue shrinkage is a well-known phenomenon, especially for subcapsular ablations, which might lead to underestimation of the ablation zone volume and therefore might influence the results of the *R*(AZV:E) [29]. Although the exact amount of tissue shrinkage remains unknown, no differences in *R*(AZV:E) were observed between subcapsular and non-subcapsular tumours.

## 5. Conclusions

In conclusion, this study confirms the unpredictability of ablation zone volumes based on the applied output energy for HCC and CRLM. No significant differences in *R*(AZV:E) were observed between CRLM and HCC. The ratio between AZV and applied energy was different between the two MWA devices, however, its reflected energy due to differences in cable and antenna design remains unclear and might contribute to these differences. Future research should focus on the delivered energy in the tissue instead of the output energy of the MWA devices, as reported by the manufacturers.

## Figures and Tables

**Figure 1 cancers-14-05570-f001:**
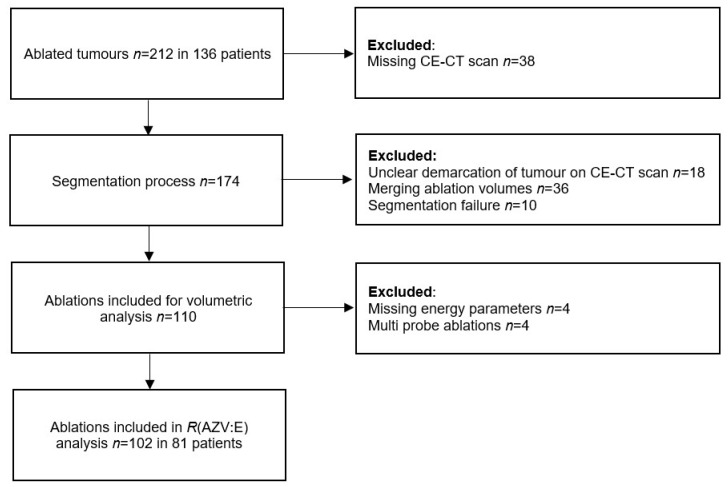
Flowchart of ablations included for volumetric analysis of ratio between ablation zone volume and applied energy, *R*(AZV:E). CE-CT: contrast-enhanced computed tomography; *R*(AZV:E): ratio between ablation zone volume and applied energy.

**Figure 2 cancers-14-05570-f002:**
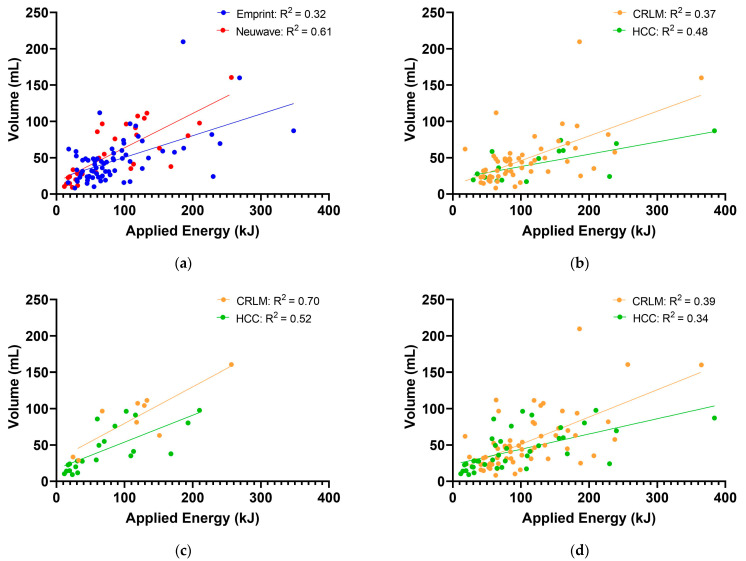
Correlation between applied energy and (**a**) ablation zone volume grouped by the two ablation devices; (**b**) ablation zone volume obtained with Emprint device grouped by tumour type; (**c**) ablation zone volume obtained with Neuwave device grouped by tumour type and (**d**) ablation zone volume grouped by the two tumour types. HCC = hepatocellular carcinoma; CRLM = colorectal liver metastasis.

**Table 1 cancers-14-05570-t001:** Clinicopathological characteristics.

**Patient Characteristics**	**Total (*n* = 81)**	**Emprint (*n* = 55)**	**Neuwave (*n* = 26)**	***p*-Value ^#^**
Sex (male:female)	45:36	26:29	19:7	**0.029**
Age in years at intervention *	66.0 (1.0)	64.3 (9.2)	69.3 (7.4)	**0.012**
BMI *	29.0 (0.7)	28.8 (6.6)	29.3 (4.0)	0.627
Liver parenchyma				
*Normal* *Steatosis* *Cirrhosis*	31 (38.3) 24 (29.6) 26 (32.1)	25 (45.5) 19 (34.5) 11 (20.0)	6 (23.1) 5 (19.2) 15 (57.7)	**0.003**
Prior systemic chemotherapy				
*No* *Neo-adjuvant for this ablation* *>6 months before*	57 (70.4) 16 (19.8) 8 (9.8)	36 (65.5) 13 (23.6) 6 (10.9)	21 (80.8) 3 (11.5) 2 (7.7)	0.353
**Tumour Characteristics**	**Total (*n* = 102)**	**Emprint (*n* = 71)**	**Neuwave (*n* = 31)**	***p*-Value ^#^**
Type of tumour				
*CRLM* *HCC*	65 (63.7) 37 (36.3)	56 (78.9) 15 (21.1)	9 (29.0) 22 (71.0)	**0.001**
Location				
*Left-sided liver*	37 (36.3))	24 (33.8)	13 (41.9)	0.504
*Right-sided liver*	65 (63.7)	47 (66.2)	18 (58.1)	
Subcapsular (<5 mm)	66 (64.7)	44	22	0.382
Perivascular (<5 mm of >3 mm-sized vessel)	30 (29.4)	24	6	0.141
Diameter (mm) at intervention †	19 (14–27)	19 (13–25)	19 (10–28)	0.743
Volume (ml) at intervention †	2.92 (1.07–7.95)	2.7 (1.0–7.8)	3.9 (1.1–9.4)	0.385

Unless otherwise specified, data are shown as number and percentages. ^#^ Reported *p* values correspond to the comparison between the two ablation devices. * Data are shown as mean with standard deviation in parentheses. † Data are shown as medians with interquartile ranges in parentheses. CRLM: colorectal liver metastasis; HCC: hepatocellular carcinoma; BMI: body mass index; The bold: *p* < 0.05.

**Table 2 cancers-14-05570-t002:** Ablation outcome parameters according to ablation device.

	Emprint (*n* = 71)	Neuwave (*n* = 31)	*p*-Value
*R*(AZV:E) (ml/kJ)	0.41 (0.33–0.56)	0.81 (0.47–0.89)	**<0.001**
Tumour diameter (mm)	19 (13–25)	19 (10–28)	0.743
Tumour volume (ml)	2.7 (1.0–7.8)	3.9 (1.1–9.4)	0.385
Ablation zone volume (ml)	41.9 (24.1–58.7)	41.1 (23.9–91.2)	0.342
Applied energy (kJ)	85.5 (60.0–156.0)	67.2 (24.0–126.0)	0.062
Needle positions (*n*)	3 (1–4)	3 (2–3)	0.108
*R*(T:AZV)	0.06 (0.04–0.15)	0.07 (0.04–0.15)	0.513

Data are shown as medians with interquartile ranges in parentheses. *R*(AZV:E): ratio between ablation zone volume and applied energy; *R*(T:AZV): ratio between tumour volume and ablation zone volume. The bold: *p* < 0.05.

**Table 3 cancers-14-05570-t003:** Ablation characteristics stratified per tumour type (HCC or CRLM) according to ablation device.

	Emprint (*n* = 71)			Neuwave (*n* = 31)		
	HCC (*n* = 15)	CRLM (*n* = 56)	*p*-Value	HCC (*n* = 22)	CRLM (*n* = 9)	*p*-Value
*R*(AZV:E) (ml/kJ)	0.38 (0.27–0.54)	0.43 (0.34–0.57)	0.398	0.79 (0.41–0.90)	0.85 (0.66–0.90)	0.384
Tumour diameter (mm)	23 (16–32)	18 (13–26)	0.073	18 (13–29)	26 (14–34)	0.337
Tumour volume (ml)	5.0 (2.2–13.0)	2.3 (0.8–7.4)	0.052	3.3 (1.1–8.3)	4.6 (1.1–9.9)	0.663
Ablation zone volume (ml)	36.1 (19.6–60.0)	42.9 (24.5–55.6)	0.866	32.2 (18.6–77.1)	96.6 (48.3–109.3)	**0.005**
Applied energy (kJ)	108.0 (57.6–162.0)	85.1 (60.8–133.7)	0.678	59.3 (23.0–109.2)	126.0 (49.8–140.1)	**0.029**
Needle positions (*n*)	3 (2–4)	3 (2–4)	0.609	3 (2–3)	2 (2–3)	0.752
*R*(T:AZV)	0.22 (0.05–0.38)	0.06 (0.03–0.13)	**0.010**	0.08 (0.05–0.21)	0.04 (0.03–0.11)	0.067

Data are shown as medians with interquartile ranges in parentheses. R(AZV:E): ratio between ablation zone volume and applied energy; *R*(T:AZV): ratio between tumour volume and ablation zone volume; The bold: *p* < 0.05.

## Data Availability

Data sharing is not applicable to this article.

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
