# Peer review of "Comparison of Two 2.45 GHz Microwave Ablation Devices with Respect to Ablation Zone Volume in Relation to Applied Energy in Patients with Malignant Liver Tumours"

_cancers, 2022, doi:10.3390/cancers14225570_

Round 1

Reviewer 1 Report

This is an interesting, well designed, and well conducted study.

I have no significant remarks about its quality, but just a perplexity on the ablation procedure. The authors reported a median tumour volume of 2.92 mL and a median number of antenna positions of 3: are not three antenna positions too many in comparison with tumour volume?

Furthermore, the study has some limits that do not reduce its overall interest, but weaken its clinical relevance, as correctly observed by the authors in their conclusions. The retrospective nature of the study does not allow to have data on the amount of reflected energy of the two devices. Consequently it is hard to draw any conclusion about the energy needed by the two devices to obtain the same ablation volume.

In my opinion the paper merits to be published as a pilot study, propaedeutic to a prospective study based on the delivered energy in the tissue instead of the output energy, preferably comparing the devices each other and with a MWA system equipped with a miniaturized device against the back heating effect (AMICA MWA System, HS Hospital Service, Aprilia, Italy).

In conclusion, my congratulations to the Authors

Author Response

Thank you for your positive feedback. Indeed, as mentioned in the conclusion: Future research should focus on the delivered energy in the tissue instead of the output energy of the MWA devices, as reported by the manufacturers.   

Reviewer 2 Report

As the authors state the first aim of the article is the comparison of two 2.45 GHz MWA devices with respect to AZV in relation to the applied energy after MWA in patients with HCC and CRLM. However, this comparison is problematic because the patient parameters are too heterogeneous between both devices. In sex, liver parenchyma and tumour type the differences are significant (table 1). In tumour volume and tumour location the differences can´t be neglected, too. Maybe, this is partly caused by the low number of the Neuwave treatments. The authors have to discuss this in detail or change the intent and aim of the article.

Investigating separately the single devices the authors found correlations between AZV and the applied energy, but they estimated it as not strong enough to predict the AZ from the applied energy in sufficient accuracy.

Therefore, the second aim of the article is the search for further parameters that influences the AZV/energy ratio.

Although the authors study different candidates, they didn´t find any such parameter, whereas they conclude that a prediction of the size of the AZV can´t be done simply on the amount of delivered energy by the device. Thus, they rightly emphasize that monitoring of the AZ during the treatment remains crucial.

Some further remarks:

1.       In line 17: Different instead of Lower.

2.       In fig.2: b shows the results of both devices, c only Emprint and d only Neuwave. In the figure text and the article text the sequence is different.

3.       In parts of results (3.3) the authors often formulate “there are no differences” or “don´t differ” whereas the values are in fact quite different but not significant (e.g. AZV(CRLM) and AZV(HCC) or E(CRLM) and E(HCC)). Therefore, they should formulate “no significant difference”.

4.       In table 3 some values are unclear: the p-value <0.001 in the first line seems to be too low, if it is the p-value for the different tumour types in Neuwave treatments.
In the last line, the HCC value 0.22 seems to be too high. It doesn´t fit to the R(T:AZV)(HCC) value in the text (0.07). These numbers have to be checked.

Author Response

Reviewer 2:

As the authors state the first aim of the article is the comparison of two 2.45 GHz MWA devices with respect to AZV in relation to the applied energy after MWA in patients with HCC and CRLM. However, this comparison is problematic because the patient parameters are too heterogeneous between both devices. In sex, liver parenchyma and tumour type the differences are significant (table 1). In tumour volume and tumour location the differences can´t be neglected, too. Maybe, this is partly caused by the low number of the Neuwave treatments. The authors have to discuss this in detail or change the intent and aim of the article.

Thank you for pointing out this important remark. We added this to the limitation section of the discussion.

Investigating separately the single devices the authors found correlations between AZV and the applied energy, but they estimated it as not strong enough to predict the AZ from the applied energy in sufficient accuracy.

Therefore, the second aim of the article is the search for further parameters that influences the AZV/energy ratio.

Although the authors study different candidates, they didn´t find any such parameter, whereas they conclude that a prediction of the size of the AZV can´t be done simply on the amount of delivered energy by the device. Thus, they rightly emphasize that monitoring of the AZ during the treatment remains crucial.

Some further remarks:

  1. In line 17: Different instead of Lower.

Changed in the manuscript.

  1. In fig.2: b shows the results of both devices, c only Emprint and d only Neuwave. In the figure text and the article text the sequence is different.

Thank you for pointing out this mistake. The figures were accidentally wrongly arranged.

  1. In parts of results (3.3) the authors often formulate “there are no differences” or “don´t differ” whereas the values are in fact quite different but not significant (e.g. AZV(CRLM) and AZV(HCC) or E(CRLM) and E(HCC)). Therefore, they should formulate “no significant difference”.

We changed this in the manuscript.

  1. In table 3 some values are unclear: the p-value <0.001 in the first line seems to be too low, if it is the p-value for the different tumour types in Neuwave treatments.
    In the last line, the HCC value 0.22 seems to be too high. It doesn´t fit to the R(T:AZV)(HCC) value in the text (0.07). These numbers have to be checked.

Thank you again for pointing out this mistake. The numbers have been checked again and changed. Besides, we have checked all the numbers, tables and figures again to prevent any further mistakes.

Reviewer 3 Report

General comment: The article presents a non-randomized retrospective study on the use of two commercially available systems for ablation of liver tumors by microwave (MW) energy. The article is generally well written and the analysis methods seem adequate. My main concern lies in the limited number of involved patients and especially in the fact that overlapping ablations were included, which prevents a direct comparison between the performance of the two systems. All these limitations must be emphasized, and the level of clinical evidence established.

Other issues:

1) The authors conclude that their results confirm the high unpredictability of AZVs based on the applied output energy. From my point of view, this statement is a bit ambiguous. For example, for the Neuwave system R2=0.61 (p<0.001), which implies that there is really a certain degree of predictability.

2) Abstract must include more information about the study type: retrospective, non randomized, etc.

3) L44-45: “The most important drawback of MWA is the appearance of viable tumour tissue at the edge of the ablation zone [ablation site recurrence (ASR)].” This statement suggests that this problem is exclusive for MWA, and not for other energy-based ablative options, such as RFA. Clarify o justify with references.

4) L48,49: “...and depends on the applied energy, controlled by the power and time setting of the 48 ablation device.” To be strict, it will also depend on heat evacuation mechanisms around the MW applicator, e.g. the blood perfusion rate at the target zone.

5) What was the criterion for choosing each MWA system in each patient? Explain.

6) L71: “(2) merging AZV of multiple tumours,” clarify if you mean that only single applications were included in the study (I guess that this was a great restriction in terms of sample size, since overlapping is common in tumor ablation to cover the entire tumor).

7) Since Ablation zone and tumour volumes were ‘volumetrically’ calculate, no information about the shape/geometry is reported. This information is surely very relevant in order to assess how AZ matches to tumor shape for each MW system. Could the authors provide some figure showing these volumes (AZ and tumor) for a representative case?

8) Consider explaining the meaning of ‘antenna positions’ the first time it appears. Is this related with the overlapping of ablation zones?

9) Table 1: All the information is a bit confusing. First, consider highlighting bold the significant differences (P<0.05). Second, indicate that the P-values correspond with the comparison between the two MW systems. I mean, I’m assuming that they don’t correspond with other comparisons, e.g. liver parenchyma, type of tumor, etc. At this regard, e.g., clarify if P=0.003 corresponds with the comparison between Neuwave and Emprint including the three liver parenchyma.

10) Correlations were better (closer strong) when ablations were distinguished between CRLM and HCC. Consider explaining the underlying physical reasons and possible clinical implications.

11) L222-L230: In the discussion, the authors are mixing two concepts (how AZV is correlated with applied energy, and how large is the AZV compared the tumor size). Split these concepts when you compare your results with those of other authors [13,14].

12) L228-230: “This very strongly suggests that liver parenchyma is affecting the evolution of the ablation zone significantly more than tumour tissue.” Explain much more what you mean with this statement.

13) Ratio Tumour / ablation zone volume ratio ranged from 0.04 to 0.22. I understand that you are over-treating the target zone, i.e. destroying healthy parenchyma. Is this right? Consider explaining the clinical implication of this.

14) In general, the part of the discussion where the results are compared with other authors is confusing. I propose that each paragraph focuses on each analyzed parameter.

15) L243-onwards: When the physical/technical reasons are provided to explain the differences between both system, considering also the tissue contraction (see: Lee J, at el. Direction of Tissue Contraction after Microwave Ablation: A Comparative Experimental Study in Ex Vivo Bovine Liver. Korean J Radiol. 2022 Jan;23(1):42-51. doi: 10.3348/kjr.2021.0134.) The authors mentioned this issue almost at the end (using the ref. [21]).

16) Finally, only when I reached line 262, I realized that the majority of ablations were performed with overlapping ablation zones. And then I agree with the authors when they state that because of the overlapping ablations, it is not possible to determine the differences in sphericity between both MWA devices. I would even go further, and reformulate most of the conclusions, since overlapping implies applying MW energy on a substrate with different properties, which can affect the reflected energy. In this sense, how different was the overlap between groups/systems? If this is not taken into account, the results might be biased (see the first general comment).

Author Response

Reviewer 3:

General comment: The article presents a non-randomized retrospective study on the use of two commercially available systems for ablation of liver tumors by microwave (MW) energy. The article is generally well written and the analysis methods seem adequate. My main concern lies in the limited number of involved patients and especially in the fact that overlapping ablations were included, which prevents a direct comparison between the performance of the two systems. All these limitations must be emphasized, and the level of clinical evidence established.

Other issues:

  • The authors conclude that their results confirm the high unpredictability of AZVs based on the applied output energy. From my point of view, this statement is a bit ambiguous. For example, for the Neuwave system R2=0.61 (p<0.001), which implies that there is really a certain degree of predictability.

Thank you pointing out this issue. We removed the word 'high' in this statement. However, the predictibility for the individual patient remains poor. For example Figure 2D, there is a wide range of ablation zone volumes for +/- 190 kJ of applied energy.

  • Abstract must include more information about the study type: retrospective, non randomized, etc.

We have added this information in the abstract.

  • L44-45: “The most important drawback of MWA is the appearance of viable tumour tissue at the edge of the ablation zone [ablation site recurrence (ASR)].” This statement suggests that this problem is exclusive for MWA, and not for other energy-based ablative options, such as RFA. Clarify o justify with references.

We have adapted this in the manuscript.

  • L48,49: “...and depends on the applied energy, controlled by the power and time setting of the 48 ablation device.” To be strict, it will also depend on heat evacuation mechanisms around the MW applicator, e.g. the blood perfusion rate at the target zone.

Thank you for poiting this out, we added 'amongst others' to this statement.

  • What was the criterion for choosing each MWA system in each patient? Explain.

The decision was only depending on operators choice. We added this in the methods section.

  • L71: “(2) merging AZV of multiple tumours,” clarify if you mean that only single applications were included in the study (I guess that this was a great restriction in terms of sample size, since overlapping is common in tumor ablation to cover the entire tumor).

We mean overlapping ablation zones of multiple tumours. For example 2 tumours located very close to each other, making it impossible to distinguish the ablation zones of the 2 tumours from each other. We have added this explanation to the Methods section.

  • Since Ablation zone and tumour volumes were ‘volumetrically’ calculate, no information about the shape/geometry is reported. This information is surely very relevant in order to assess how AZ matches to tumor shape for each MW system. Could the authors provide some figure showing these volumes (AZ and tumor) for a representative case?

The majority of the ablation procedures (86.3%) were performed with multiple overlapping ablations, making it impossible to adequately analyze any differences in sphericity between the ablation devices. This was mentioned in the limitation part of the discussion (line 283).

  • Consider explaining the meaning of ‘antenna positions’ the first time it appears. Is this related with the overlapping of ablation zones?

We added this in the methods section 2.2, line 92.

  • Table 1: All the information is a bit confusing. First, consider highlighting bold the significant differences (P<0.05). Second, indicate that the P-values correspond with the comparison between the two MW systems. I mean, I’m assuming that they don’t correspond with other comparisons, e.g. liver parenchyma, type of tumor, etc. At this regard, e.g., clarify if P=0.003 corresponds with the comparison between Neuwave and Emprint including the three liver parenchyma.

Significant differences are now in bold in the revised version of the manuscript.

We have added an explanation as footnote to Table 1. In addition, P=0.003 indeed corresponds with the comparison between the two ablation devices including the three types of underlying liver disease.

  • Correlations were better (closer strong) when ablations were distinguished between CRLM and HCC. Consider explaining the underlying physical reasons and possible clinical implications.

Explained variability (R2) of CRLM and HCC were very close (R2 0.39 vs 0.34) for both devices. R2 was higher for CRLM than HCC when using Neuwave (0.70 vs 0.52). However, no clear explanation for these differences could be found based on the data in the current study. We suggested some explainatin for this in the discussion (line 248).

  • L222-L230: In the discussion, the authors are mixing two concepts (how AZV is correlated with applied energy, and how large is the AZV compared the tumor size). Split these concepts when you compare your results with those of other authors [13,14].

We agree with the reviewer that it concerns two different concepts. Therefore, in the second paragraph of the Discussion, we focused on the comparison of our results regarding the relationship between AZV and applied energy with similar comparisons reported in the literature. In the third paragraph of the discussion, we described in detail the results regarding tumor size and AZV. As it concerns two separate paragraphs, we believe there is no clear mix of the two concepts.

  • L228-230: “This very strongly suggests that liver parenchyma is affecting the evolution of the ablation zone significantly more than tumour tissue.” Explain much more what you mean with this statement.

We clarified this more in detail in the discussion, and added a reference to a study reporting a similar concept (Ref 25).

  • Ratio Tumour / ablation zone volume ratio ranged from 0.04 to 0.22. I understand that you are over-treating the target zone, i.e. destroying healthy parenchyma. Is this right? Consider explaining the clinical implication of this.

This low ratio can be explained by the fact that the volume of a spherical tumour of 10 mm (tumour volume of 1 ml) would only be 7% of the AZV when aiming to obtain a tumor-free margin of 10 mm (AZV 14 ml). We clarified this in the discussion (line 242).

  • In general, the part of the discussion where the results are compared with other authors is confusing. I propose that each paragraph focuses on each analyzed parameter.

Thank you for pointing out this. We revised the discussion taking your comment into account.

  • L243-onwards: When the physical/technical reasons are provided to explain the differences between both system, considering also the tissue contraction (see: Lee J, at el. Direction of Tissue Contraction after Microwave Ablation: A Comparative Experimental Study in Ex Vivo Bovine Liver. Korean J Radiol. 2022 Jan;23(1):42-51. doi: 10.3348/kjr.2021.0134.) The authors mentioned this issue almost at the end (using the ref. [21]).

Thank you for this suggestion. We added this reference to the discussion.

  • Finally, only when I reached line 262, I realized that the majority of ablations were performed with overlapping ablation zones. And then I agree with the authors when they state that because of the overlapping ablations, it is not possible to determine the differences in sphericity between both MWA devices. I would even go further, and reformulate most of the conclusions, since overlapping implies applying MW energy on a substrate with different properties, which can affect the reflected energy. In this sense, how different was the overlap between groups/systems? If this is not taken into account, the results might be biased (see the first general comment)

See section 3.1. Only 13.7% of ablations were performed with a single antenna position. All other ablations (86.3%) were overlapping ablations. No significant differences in antenna positions were seen between the Emprint/Neuwave (p=0.108) and between tumour types (p=0.609 for Emprint and p=0.752 for Neuwave). We also described this limitation in the limitation section of the Discussion.

Round 2

Reviewer 2 Report

Only 3 remarks:

In line 24:  "or" instead of "and".

In line 180: no "of".

In line 209: a "significantly" is missing in front of different.

Reviewer 3 Report

The authors suitably addressed all my comment and concerns